

# Temperature-dependent diffusion coefficient of H$_2$SO$_4$ in air:
# laboratory measurements using laminar flow technique
David Brus[1, 2], Lenka Skrabalova [2, 3], Erik Herrmann[4], Tinja Olenius[5*], Tereza Travnickova[6] and
Joonas Merikanto[1]
[1] {Finnish Meteorological Institute, Erik Palménin aukio 1, P.O. Box 503, FIN-00100
Helsinki, Finland}
[2] {Laboratory of Aerosols Chemistry and Physics, Institute of Chemical Process Fundamentals
Academy of Sciences of the Czech Republic, Rozvojova 135, CZ-165 02 Prague 6, Czech
Republic}
[3] {Department of Physical Chemistry, Faculty of Science, Charles University in Prague,
Hlavova 8, Prague, 128 43, Czech Republic}
[4] {Laboratory of Atmospheric Chemistry, Paul Scherrer Institute, CH-5232 Villigen PSI,
Switzerland}
[5] {Department of Physics, University of Helsinki, Gustaf Hällströmin katu 2 A, P.O. Box 64,
FIN-00014 Helsinki, Finland
∗ Current address: Department of Environmental Science and Analytical Chemistry (ACES) and
Bolin Centre for Climate Research, Stockholm University, SE-10691 Stockholm, Sweden}
[6] {Department of Multiphase Reactors, Institute of Chemical Process Fundamentals Academy
of Sciences of the Czech Republic, Rozvojova 135, CZ-165 02 Prague 6, Czech Republic}
Correspondence to: D.Brus (david.brus@fmi.fi)





**Abstract**
We report measurements of the diffusion coefficient of sulfuric acid in humidified air at a range
of relative humidities (from ~4 to 70 %), temperatures (278, 288 and 298 K) and initial $H_2SO_4$
concentration (from $1 \times 10^6$ to $1 \times 10^8$ molec. cm$^{-3}$). The diffusion coefficients were estimated from
the sulfuric acid wall loss rate coefficients under laminar flow conditions. The flow conditions
were verified with additional fluid dynamics model CFD-FLUENT simulations which also
reproduced the loss rate coefficients very well at all three temperatures with the maximum
difference of 7 % between the measured and simulated values. The concentration of $H_2SO_4$ was
measured continuously with chemical ionization mass spectrometer (CIMS) at seven different
positions along the flow tube. The wall losses of $H_2SO_4$ were determined from the slopes of fits
to measured $H_2SO_4$ concentrations as a function of the position along the flow tube. The
observed wall loss rate coefficients, and hence the diffusion coefficients, were independent of
different initial $H_2SO_4$ concentrations and different total flow rates. However, the determined
diffusion coefficients decreased with increasing relative humidity, as also seen in previous
experiments, and had a rather strong power dependence of the diffusion coefficient with respect
to temperature, around $\propto T^{5.4}$, which is in disagreement with the expected temperature
dependency of $\sim T^{1.75}$ observed for other gases and not tested before for sulfuric acid. The effect
of relative humidity on the diffusion coefficient is likely due to stronger hydration of $H_2SO_4$
molecules and likely also due to the presence of trace impurities such as amines, possibly
brought to the system by humidification. Clustering kinetics simulations using quantum chemical
data suggest that also the strong temperature dependence of the observed diffusion coefficient
might be explained by increased diffusion volume of $H_2SO_4$ molecules due to stronger clustering
with base-impurities like amines.
**1. Introduction**
Sulfate aerosols play a dominant role in atmospheric chemistry and have undoubtedly influence
on humans' health and Earth's climate. Gas phase sulfuric acid is formed via oxidation reaction
of $SO_2$ with OH radicals. The loss of gaseous $H_2SO_4$ in the atmosphere is caused by new particle
formation events, acid-base reactions and cluster formation, and condensation on pre-existing
atmospheric particles. The growth of a particle is driven by condensation of surrounding vapour
on its surface, and the diffusion coefficient ($D$) of $H_2SO_4$ is often used in mass transport
calculations in aerosol chemistry and physics. Condensation and evaporation rates are key
parameters in aerosol dynamics models, and the accuracy of these rates is highly dependent on





the values used for the binary diffusion coefficients (Seinfeld and Pandis, 1998). Under certain
circumstances, the gas phase diffusion can even limit the overall rates of condensation and
reactions of trace gases with aerosol particles via influencing the uptake of gas molecules onto
the surface (Tang et al., 2014). The factor that determines if a $H_2SO_4$ molecule will attach to pre-
existing aerosol or stay in the gas phase, possibly contributing to subsequent new particle
formation, is the mass accommodation coefficient (Pöschl et al., 1998). Together with
information on the mass accommodation coefficient, detailed knowledge on the diffusion
coefficient is necessary for accurately simulating atmospheric condensation processes.
In this paper we report laboratory measurements of the diffusion coefficient of sulfuric acid in
air. The diffusion coefficient of $H_2SO_4$ was estimated from the first order rate coefficients of the
wall losses of $H_2SO_4$ in a flow tube. The measurements were conducted at atmospheric pressure
under different experimental conditions in order to assess the effect of temperature, relative
humidity, residence time and initial $H_2SO_4$ concentration on the diffusion coefficient of $H_2SO_4$.
All previous measurements of the sulfuric acid diffusion coefficient have been carried out using
nitrogen as the carrier gas and a laminar flow technique. Pöschl et al. (1998) studied the gas-
phase diffusion of $H_2SO_4$ at $T$=303 K, Lovejoy and Hanson (1996) performed experiments at
$T$=295 K, and Hanson and Eisele (2000) at $T$=298 K. To our best knowledge, we here present the
first study that investigates systematically the temperature dependency of the diffusion
coefficient of $H_2SO_4$. In a previous study (Hanson and Eisele, 2000) the RH dependency of
$H_2SO_4$ diffusion was investigated, but results reporting the temperature dependency have not
been published before.
The Chapman-Enskog theory on gas kinetics predicts the binary diffusion coefficient to depend
on the temperature as $D \propto T^{1.5}$ when approximating the gas molecules as hard spheres. Fuller et
al. (1966) used a semi-empirical method based on the best nonlinear least square fit for a
compilation of 340 experimental diffusion coefficients, and obtained a temperature dependence
of $T^{1.75}$. The Fuller et al. method is known to yield the smallest average error, hence it is still
recommended for use (Reid et al., 1987). According to a compilation work of Marrero and
Mason (1972), the temperature dependence of diffusion coefficients in binary gas mixtures in
most cases varies between $T^{1.5}$ and $T^2$. However, gaseous sulfuric acid vapour can undergo strong
clustering due to presence of base impurities, as noted in several previous experiments (e.g.
Petäjä et al 2011; Almeida et al., 2013; Neitola et al. 2015, Rondo et al. 2016). Such base
impurities are unavoidably present also in our experiment, and most probably they originate



from the humidification of the carrier gas (e.g. Benson et al., 2011; Kirkby et al., 2011; Neitola
et al. 2015). Cluster kinetic simulations have suggested that the diffusion coefficient of sulfuric
acid is likely sensitive to such clustering (Olenius et al., 2014), which on the other hand is
sensitive to temperature. Here, we use several approaches to verify the experimental method, and
examine our results against the predictions of the semi-empirical formula as well as data from
the previous experiments. In addition, we assess the effect of molecular cluster formation by
cluster kinetics simulations with quantum chemical input data.

## 2. Methods

The experimental setup used in this study was described in detail in our previous work (Neitola
et al., 2014; Skrabalova et al., 2014) and therefore only a brief description is given here. The
whole experimental apparatus consists of four main parts: a saturator, a mixing unit, a flow tube
and the sulfuric acid detection system – Chemical Ionization Mass Spectrometer (CIMS) (Eisele
and Tanner, 1999), presented in Figure 1. The $H_2SO_4$ wall loss measurements were carried out in
a laminar flow tube at three temperatures of 278.20(±0.2), 288.79(±0.2), and 298.2(±0.2) K
using purified, particle free and dry air as a carrier gas. The flow tube is a vertically mounted
cylindrical tube made of stainless steel with an inner diameter (I.D.) of 6 cm and a total length of
200 cm. The whole flow tube was insulated and kept at a constant temperature with two liquid
circulating baths (Lauda RK-20). The flow tube consists of two 1 meter long parts; one of them
is equipped with 4 holes in the distance of 20 cm from each other, see Figure 1. Sulfuric acid
vapour was produced by passing a stream of carrier gas through a saturator filled with 95-97 %
wt. sulfuric acid (J.T. Baker analysed). As a saturator we used a horizontal iron cylinder with
Teflon insert (I.D. 5 cm) and it was thermally controlled using a liquid circulating bath (LAUDA
RC-6). The temperature inside the saturator was measured with a PT100 thermocouple (± 0.05
K). The carrier gas saturated with $H_2SO_4$ was then introduced with a flow rate from 0.1 to 1 l
$min^{-1}$ into the mixing unit made of Teflon and turbulently mixed with a stream of humidified
particle free air. The gas mixture was then introduced into the flow tube. The flow rate of the
mixing air varied in most of the experiments from about 7 to 10 l $min^{-1}$. The mixing air was
humidified with one pair of Nafion humidifiers (Perma Pure MH-110-12) connected in parallel,
where the flow of the mixing air was split into half for longer residence time and better
humidification in both humidifiers. Ultrapure water (Millipore, TOC less than 10 ppb, resistivity
18.2 MΩ.cm @25°C) circulating in both humidifiers was temperature controlled with liquid
circulating bath (Lauda RC-6 CS). The mixing unit was kept at room temperature and it was not



insulated. The mixing unit had following dimensions: O.D. = 10 cm, I.D. = 7 cm and height = 6
cm. Both lines of the carrier gas (saturator and mixing air) were controlled by a mass flow
controller to within ±3 % (MKS type 250). The relative humidity was measured at the centre and
far end of the flow tube with two humidity and temperature probes (Vaisala HMP37E and
humidity data processor Vaisala HMI38) within accuracy of ± 3 %.
The sulfuric acid diffusion coefficients were estimated as a function of relative humidity from
the $H_2SO_4$ loss measured by CIMS along the flow tube. The detailed information regarding the
operational principles and calibration of CIMS is given in Eisele and Tanner, (1993); Mauldin et
al., (1998) and Petäjä et al., (2009) and therefore will not be given here again. The charging and
detection efficiency of CIMS in the presence of trace concentrations of base impurities is
discussed thoroughly theoretically (e.g. Kupiainen-Määttä et al., 2013; Ortega et al., 2014) and
also in recent experimental reports (e.g. Neitola et al., 2015; Rondo et al., 2016). Possible
attachment of base and/or water molecules to single $H_2SO_4$ molecules is not expected to have a
notable effect on their detection efficiency. However, both free $H_2SO_4$ molecules and those
bound to base and/or water molecules are detected as single $H_2SO_4$ molecules by CIMS, since
the ligands are quickly lost upon the chemical ionization (e.g. Ortega et al., 2014). In this study
the actual $H_2SO_4$ concentrations are not of particular interest, we focus here only on relative loss
of $H_2SO_4$ along the flow tube. The concentration of sulfuric acid in gas phase was measured as
97 m/z Da using CIMS along the flow tube (see Fig. 1) at the beginning (0 cm), in the middle
(100 cm) and at the lower part in distances of 120, 140, 160, and 180 cm from the beginning and
at the outlet (200 cm) of the flow tube in a wide range of relative humidities from 4 to 70 %. The
CIMS sampling flow rate was set to 7 l min$^{-1}$. In order to measure the $H_2SO_4$ concentration along
the whole flow tube, an additional CIMS inlet sampling tube was used - a stainless steel tube
with I.D. 10 mm and whole length of 122 cm (100 cm straight + 22 cm elbow-pipe). The
experimental measurement proceeded in the following way. First, all the experimental conditions
(temperature of saturator and flow tube, flow rates, relative humidity) were adjusted. When the
steady state was reached, the CIMS' inlet was connected to the lowest hole at 200 cm and
concentration of sulfuric acid was recorded for at least 20 min. Afterwards the CIMS' inlet was
moved up to the hole at 180 cm, and the same procedure was repeated until the last hole at the
top of the tube was reached. To confirm the reproducibility of the experimental data the $H_2SO_4$
concentration at any arbitrary distance along the flow tube was measured again. Moreover, the
reproducibility was checked by exchanging the flow tube parts, so that the part with 4 holes was



moved up, and $H_2SO_4$ losses were measured in the distances 0, 20, 40, 60, 80, 100 and 200 cm,
respectively.

## 2.1  The CFD model

To verify the assumption of laminar flow inside the tube, we applied the computational fluid
dynamics model CFD-FLUENT (version 6.2) which simulates flow based on the Euler equations
for mass and momentum conservation. These equations and the general operating principles of
FLUENT are described in detail in Herrmann et al. (2006, 2009 and 2010). It has to be noted that
unlike the earlier studies, this work did not include the Fine Particle Model (FPM). Particle
production was thus not taken into account, and only sulfuric acid and water vapours are
considered in the CFD-FLUENT simulations.
The simulations only considered the flow tube part of the experimental setup described in section
2. Methods; the flow tube can be set up as an axisymmetric 2D problem. For the calculations
presented here, we chose a resolution of $50 \times 1000$ cells. The same geometry has been used in
Herrmann et al. (2010). Boundary conditions (volumetric flow, wall temperatures, relative
humidity, and initial sulfuric acid concentration) were set to match the experimental conditions.
The wall was assumed to be an infinite sink for sulfuric acid, which means that the $H_2SO_4$
concentration at the walls was set to 0 in the simulations. Properties of sulfuric acid were
identical to the ones described in our earlier work. Differing from Herrmann et al. (2010), there
is no temperature gradient or buoyancy phenomena disturbing parabolic radial flow profile. To
verify the proper operation of the setup we applied the diffusion coefficient derived
experimentally in this work to FLUENT simulations. The simulations yielded a profile of
sulfuric acid concentration inside the flow tube which we compared back to the experimental
results.

## 2.2  Experimental determination of the diffusion coefficient

The wall loss of $H_2SO_4$ in the flow tube was assumed to be a diffusion controlled first-order rate
process, which can be described by a simple equation:
$$[H_2SO_4]_t = [H_2SO_4]_0 \, e^{-kt}, \tag{1}$$
where $[H_2SO_4]_0$ is the initial concentration of $H_2SO_4$, $[H_2SO_4]_t$ is the concentration after time $t$
and $k$ $(s^{-1})$ is the rate constant, which is given by the equation:



$$k = 3.65 \frac{D}{r^2},\qquad\qquad\qquad (2)$$
where $r$ is the radius of the flow tube and $D$ is the diffusion coefficient of $H_2SO_4$. Equation 2 is
valid for diffusion in a cylindrical tube under laminar flow conditions and when the axial
diffusion of the species investigated is negligible (Brown, 1978). The slopes obtained from linear
fits to the experimental data $\ln([H_2SO_4])$ as a function of the distance in the flow tube stand for
the loss rate coefficient, $k_{obs}$ (cm$^{-1}$) assuming that the first order loss to the flow tube wall is the
only sink for the gas phase $H_2SO_4$. Multiplying the loss rate coefficient $k_{obs}$ with mean flow
velocity in the flow tube (cm s$^{-1}$) yields the experimental first-order wall loss rate coefficient $k_w$
(s$^{-1}$), from which the diffusion coefficients of $H_2SO_4$ were determined using Eq. 2. Hanson and
Eisele (2000) reported that the wall of the flow tube can act as a source of $H_2SO_4$ vapour after
exposure in long lasting experiments and under very low relative humidity (RH $\leq$ ~0.5%).
The accuracy of our RH measurements is $\pm$ 3 % RH, so to avoid any influence of $H_2SO_4$
evaporation from the flow tube wall we only used data measured at RH $\geq$ 4 % in the final
analysis. Furthermore, we performed CFD-FLUENT simulations at RH=5 % and $T$=298 K, with
increased $H_2SO_4$ concentration on the flow tube wall (0-100 % of $[H_2SO_4]_0$), shown in Fig. 2.
The comparison suggests that in our measurements the concentration on the flow tube wall is
below 6 % of the $[H_2SO_4]_0$ under all conditions. When the $H_2SO_4$ concentration on the wall is $\leq$ 6
% of $[H_2SO_4]_0$, the resulting difference in the obtained diffusion coefficient is within 10 % when
compared to diffusion coefficient obtained with infinite sink boundary condition on the wall, as
indicated by the shaded box in bottom left corner in Fig 2B. Any higher $H_2SO_4$ concentration at
the wall than 6 % of $[H_2SO_4]_0$ would lead to a larger than 10 % decrease in the obtained diffusion
coefficient.
**2.3 Quantum chemical data and cluster kinetics modeling**
To assess the effects of possible base impurities on the measurement results, we performed
clustering kinetics simulations using quantum chemical input data for the stabilities of $H_2SO_4$–
base clusters as described by Olenius et al. (2014). Since recent theoretical studies (e.g. Ortega et
al., 2012; Kupiainen-Määttä et al., 2013; Loukonen et al. 2014) suggest and experiments (e.g.
Zollner et al. 2012; Almeida et al. 2013; Kürten et al., 2014; Neitola et al., 2014, Rondo et al.
2016) confirm that amines are more effective in stabilizing sulfuric acid clusters than ammonia,
we focus only on the clustering of sulfuric acid with dimethylamine (DMA) and trimethylamine
(TMA) and their hydrates (see also Section 3.2 Effect of Base Contaminants in Olenius et al.,





2014).The cluster kinetics approach does not consider the 2D flow profile, but only the central
stream line of the flow, from which clusters and molecules are lost by diffusion. This is
considered a reasonable assumption for a laminar flow, as also indicated by the CFD-FLUENT
modeling results (see the Results section).
Detailed information on the simulations, as well as extensive discussion on the effects of
clustering on the apparent diffusion coefficient can be found in the study by Olenius et al.
(2014). Theoretical diffusion coefficients for sulfuric acid and representative base contaminant
molecules and small acid–base clusters and their hydrates were calculated according to the
kinetic gas theory. The effective diffusion coefficient corresponding to the experimental
approach was determined by simulating the time evolution of the molecular cluster
concentrations using cluster evaporation rates based on quantum chemical calculations at the
B3LYP/CBSB7//RICC2/aug-cc-pV(T+d)Z level of theory, as described by Olenius et al. (2014).
The simulations were run by setting initial concentrations for $H_2SO_4$ and base monomers, and
integrating the time development of the cluster concentrations for the experimental residence
times. The initial acid concentration was set to be the average $H_2SO_4$ monomer concentration
$[H_2SO_4]= 5\times10^6$ cm$^{-3}$ measured with CIMS at the beginning of the flow tube (see Fig. 1, hole 1)
for all experimental conditions. For the initial base concentration we adopted a similar approach
as described in Olenius et al. (2014): The initial base concentration was considered to be a)
constant during the experiment, or b) RH dependent, i.e. base molecules enter the system with
the water vapour; such a scenario seems to be reasonable since it was observed in several
experimental set-ups (e.g. Benson et al., 2011; Brus et al., 2011; Kirkby et al., 2011). In the
second case we set the initial base concentration $[base]_{init}$ to be linearly proportional to RH as

$[base]_{init}$ (ppt) = $[base]_{dry}$ (ppt) + $0.1\times$ RH (%),               (3)

where the linear relationship was based on a fit to DMA and TMA concentrations measured in
the same experimental setup, but different experiments at various RH and $H_2SO_4$ concentrations
(Neitola et al., 2014 and 2015). The dry values $[base]_{dry}$ were taken from Brus et al., (2016). The
resulting initial base concentrations $[base]_{init}$ for DMA and TMA were 4 and 2 ppt, respectively,
at dry conditions (RH=0 %), and 10 and 8 ppt at RH=60 %. The simulations were performed for
the three temperatures of 278, 288 and 298 K at atmospheric pressure (1 atm). The temperature
dependency of the viscosity of the carrier gas, needed to calculate the diffusion coefficients of
different species in the simulations (see Eq. (6) in Olenius et al., 2014), was taken to be




$\qquad \eta_{N_2} = \eta_{N_2,0} \left(\frac{T_0 + C}{T + C}\right) \left(\frac{T}{T_0}\right)^{3/2},$ (4)
where $\eta_{N_2,0} = 17.81 \times 10^{-6}$ Pa s, $T_0 = 300.55$ K, and $C = 111$ K (Crane Co., 1982).
**3. Results and discussion**
Figure 3 shows the diffusion coefficients of $H_2SO_4$, determined from the loss rate coefficients $k_w$
($s^{-1}$) using Eq. 2 as a function of RH at the three temperatures of 278, 288 and 298 K. The
measured points are accompanied with the fit and $H_2SO_4$ - $N_2$ data at 298 K reported by Hanson
and Eisele (2000). As can be seen, the diffusion coefficient values decreased as the RH was
increased and the diffusion coefficient dependency on RH flattens in the range of RH 20-70 %.
These results show lower wall losses and slower diffusion to the wall due to strong hydration of
$H_2SO_4$ molecules (Jaecker-Voirol and Mirabel, 1988) and possibly $H_2SO_4$ clustering with base
impurities.
There are three previously reported experimental values of the $H_2SO_4$ diffusion coefficient in
nitrogen. Pöschl et al. (1998) reported a value of 0.088 $cm^2$ $s^{-1}$ at $T$=303 K and RH $\leq$ 3%,
Lovejoy and Hanson (1996) reported a value of 0.11 $cm^2$ $s^{-1}$ at $T$=295 K and RH $\leq$ 1%, and the
study of Hanson and Eisele (2000) yielded a value of 0.094 $cm^2$ $s^{-1}$ at $T$=298 K and RH $\leq$ 1%.
The value of the diffusion coefficient of $H_2SO_4$ in air at $T$=298 K and RH=4 % determined in this
study is 0.08 $cm^2 s^{-1}$, which is in reasonable agreement with previously reported values, although
the comparison is complicated because of slightly different experimental conditions and different
carrier gases.
In Fig. 4 the $H_2SO_4$ losses simulated with the CFD-FLUENT model described in section 2.1 are
compared with experimental values, which were measured in a separate set of experiments
conducted at $T$=278, 288 and 298 K. The linear fit to the experimental data represents the loss
rate coefficients ($k_{obs}$, $cm^{-1}$). As can be seen from Fig. 4, the model describes the behaviour of
$H_2SO_4$ in the flow tube very well and confirms the validity of laminar flow approximation for all
three temperatures. The maximum difference between the experimental and simulated values of
the loss rate coefficient ($k_{obs}$) was found 7 %, see Fig. 4D for details.
Semi-empirical predictions for binary diffusion coefficients can be calculated from the Fuller et
al. equation which is based on fits to experimental data of various gases as described by Fuller et
al. (1966) and Reid at al. (1987):

$\qquad D_{AB} = \dfrac{0.00143 T^{1.75}}{P \sqrt{M_{AB}} \times \left[\sqrt[3]{(\Sigma_v)_A} + \sqrt[3]{(\Sigma_v)_B}\right]^2},$ (4)




where $D_{AB}$ is the binary diffusion coefficient of species A and B (cm$^2$ s$^{-1}$), $T$ is the temperature
(K), $P$ is the pressure (bar), $M_{AB}$ is $2[(1/M_A) + (1/M_B)]^{-1}$ (g mol$^{-1}$), where $M_A$ and $M_B$ are the
molecular weights of species A and B (g mol$^{-1}$), and $\sum_v$ is calculated for each component by
summing its atomic diffusion volumes (Reid at al., 1987). The functional form of Eq. (4) is
based on the kinetic gas theory (the Chapman-Enskog theory), and the temperature dependence
is obtained from a fit to a large set of experimental diffusion coefficients. A purely theoretical
approach based on the kinetic gas theory with the hard-spheres approximation would yield a
dependence of $T^{1.5}$. The calculated values of the diffusion coefficients of H$_2$SO$_4$, dimethylamine-
and trimethylamine-sulfate in dry air at 298 K using the Fuller method are 0.11, 0.08 and 0.074
cm$^2$ s$^{-1}$, respectively, which is in a reasonable agreement with our experimental data – the
measured diffusion coefficient of H$_2$SO$_4$ at $T$=298 K under close to dry conditions (RH 4%) is
0.08 (cm$^2$ s$^{-1}$). However, when calculating the diffusion coefficients of H$_2$SO$_4$ in dry air at lower
temperatures (278 and 288 K) with the Fuller method, the agreement of the experimental values
with the predictions deteriorates. The formula predicts significantly higher diffusion coefficients
than those observed in the experiments. The calculated values of $D_{AB}$ for H$_2$SO$_4$ are 0.104 cm$^2$ s$^{-}$
$^1$ at $T$= 288 K and 0.098 cm$^2$ s$^{-1}$ at $T$=278 K, and the measured values are 0.07 cm$^2$ s$^{-1}$ at ($T$=288
K and RH=8 %) and 0.054 cm$^2$ s$^{-1}$ at ($T$=278 K and RH=16 %), respectively. The temperature
dependency of the experimental diffusion coefficients was found to be a power of 5.4 for the
whole dataset and temperature range. Since the data show a clear stepwise temperature
dependency we provide also two separate fits to data from 278 to 288 K and from 288 to 298 K,
with power dependencies of 2.2 and 8.7, respectively.  These numbers are striking when
compared to the empirical method of Fuller et al. (1966), who obtained the best fit to 340
experimental diffusion coefficients with the power dependency of $T \propto 1.75$.
In Figure 5 we show the temperature dependency of the experimental data obtained from
literature (Lovejoy and Hanson, 1996; Pöschl et al., 1998; Hanson and Eisele, 2000), predictions
of the Fuller method for the diffusion coefficients of sulfuric acid, dimethylamine- and
trimethylamine-sulfates in dry air, results of the clustering kinetics simulations using quantum
chemical data for several simulated systems (discussed below), and the experimental data of this
work. The data collected from literature, all obtained using laminar flow technique and N$_2$ as the
carrier gas, show a temperature dependency opposite to the one expected from theory. However,





the range of temperatures at which the measurements were carried out is quite narrow (only 8 K)
and different experimental set-ups could explain such behaviour.
The origin of the discrepancy in the temperature dependency of the diffusion coefficient in our
experiment remains unclear; however, a possible explanation could be the increased clustering of
$H_2SO_4$ at lower temperatures (see explanation below) with unavoidably present trace impurities
in the system, such as amines. The CIMS was used to measure the concentrations of $H_2SO_4$ gas
phase monomers and dimers during the experiments; larger clusters were outside the mass range
of the CIMS used. In order to explain the experimental observation of the temperature
dependency of the diffusion coefficient, the dimer to monomer ratio at different temperatures
was investigated. The $H_2SO_4$ dimer formation is a result of $H_2SO_4$ monomer collisions, and thus
the observed $H_2SO_4$ dimer CIMS signal depends on the $H_2SO_4$ monomer concentration and also
on the residence time, which determinates the time available for the clustering to take place
(Petäjä et al., 2011). No significant temperature dependency of the [dimer]/[monomer] ratio was
observed in our experiments, which is in agreement with Eisele and Hanson (2000), who report a
relatively constant $H_2SO_4$ [dimer]/[monomer] ratio with lowering temperature (the temperature
range investigated in their study was 235 – 250 K). On the other hand, they report a substantial
increase in the larger clusters' (trimer and tetramer) concentration with decreasing temperature
while the monomer concentration was almost constant. There are only a very few previously
reported values of the sulfuric acid dimer to monomer ratio from laboratory experiments. Petäjä
et al. (2011) studied the close to collision-limited sulfuric acid dimer formation under
experimental conditions similar to our study ($T$=293 K, RH=22 %, initial $H_2SO_4$ concentrations
from $10^6$ to $10^8$ molecule cm$^{-3}$ with saturator containing liquid $H_2SO_4$ and in-situ $H_2SO_4$ using
$O_3$-photolysis as methods for producing gas phase $H_2SO_4$). They reported $H_2SO_4$
[dimer]/[monomer] ratios ranging from 0.05 to 0.1 at RH = 22 % and a residence time of 32 s.
Petäjä  et al. (2011) speculate about the presence of a third stabilizing compound, and their
experimental dimer formation rates correspond well to modelled rates at a DMA concentration
of about 5 ppt. Almeida et al. (2013) reported [dimer]/[monomer] ratios from 0.01 to 0.06 for the
experiments in CLOUD chamber with addition of DMA (3-140 ppt, with the effect saturated for
addition >5 ppt) and [dimer]/[monomer] ratios from $1\times10^{-4}$ to 0.003 for pure binary $H_2SO_4$-
water system, both at RH=38 % and $T$=278 K. In our measurements the $H_2SO_4$
[dimer]/[monomer] ratio under conditions $T$=298 K, RH=24 % and a residence time of ~37 s,
spans the range from 0.03 to 0.11, which is in reasonable agreement with values reported by both



Petäjä et al. (2011) and Almeida et al. (2013) when trace impurity levels of DMA are present in
the system.
The cluster population simulations using quantum chemical data (see Figs. 5, 6 and Table 1)
show that the presence of base impurities may decrease the effective $H_2SO_4$ diffusion coefficient
by attachment of base molecules to the acid. Simulations considering only hydrated $H_2SO_4$
molecules and no bases give higher values for the diffusion coefficient, and also a notably less
steep temperature dependency (Table 1). Similarly to experiment the stepwise behaviour of
temperature dependency could be found when fits are performed separately for temperatures
278-288 K and 288-298 K, see Table 1. This demonstrates that temperature-dependent clustering
may change the behaviour of the effective diffusion coefficient with respect to temperature.
Results obtained by simulating clusters containing $H_2SO_4$ and DMA are closer to the
experimental diffusion coefficient values than those obtained using $H_2SO_4$ and TMA. On the
other hand, the power dependency shows a better agreement for the $H_2SO_4$–TMA system (see
Table 1).The best agreement between the simulations and the experiment was found for the
temperature 298 K. Also, changing the base concentration according to Eq. 3 shows a better
performance than keeping the base concentration constant. Allowing the formation of clusters
containing up to two $H_2SO_4$ and two base molecules has no significant effect. In principle, the
larger clusters bind $H_2SO_4$ molecules and may thus increase the apparent diffusion coefficient,
but here their effect is minor due to the relatively low initial $H_2SO_4$ concertation of $5\times10^6$ cm$^{-3}$
used in the simulations. More analysis on the effects of the amines can be found in the work by
Olenius et al. (2014).
The formation of particles inside the flow tube during the experiments was measured regularly
using Ultrafine Condensation Particle Counter (UCPC model 3776, TSI Inc. USA) with the
lower detection limit of 3 nm. The highest determined concentration of particles was
approximately $2\times10^4$ yielding the maximum nucleation rate $J$ of ~ 500 particles cm$^3$ s$^{-1}$ at
$T$=278 K and RH=60 % . Since the nucleation rate was increasing with decreasing temperature
and elevated RH in the flow tube, the loss of gas phase sulfuric acid to the particles was more
pronounced at temperatures of 288 and 278 K. The losses of $H_2SO_4$ to particles were minimal –
units of percent (see e.g. Brus et al. 2011 and Neitola et al, 2015 for details) and cannot explain
our experimental observation of increased $H_2SO_4$ diffusion coefficient temperature dependency.
The additional losses of $H_2SO_4$ would lead to increased values of observed loss rate coefficient
($k_{obs}$) and subsequently to higher diffusion coefficient.



## 4. Conclusions

We have presented measurements of sulfuric acid diffusion coefficient in air derived from the first-order rate coefficients of wall loss of $H_2SO_4$. The experiments were performed in a laminar flow tube at temperatures 278, 288 and 298 K, relative humidities from 4 to 70 %, under atmospheric pressure and at initial $H_2SO_4$ concentrations from $10^6$ to $10^8$ molec. $cm^{-3}$. The chemical ionization mass spectrometer (CIMS) was used to measure $H_2SO_4$ gas phase concentration at seven different positions along the flow tube. The wall losses were determined from the linear fits to experimental $\ln[H_2SO_4]$ as a function of axial distance in the flow tube. The losses of $H_2SO_4$ inside the flow tube were also simulated using a computational fluid dynamics model (CFD-FLUENT), in which the wall is assumed to be an infinite sink for $H_2SO_4$. The experimentally determined $H_2SO_4$ losses along the flow tube were in a very good agreement with profiles calculated using the FLUENT model, where experimentally obtained diffusion coefficients were used as an input. A maximum difference of ~7 % for experiments conducted at $T$=278, 288 and 298 K and in the whole RH range was found when compared to model. The results of the fluid dynamics model (CFD-FLUENT) also satisfactory confirm the assumption of fully developed laminar profile inside the flow tube and infinite sink boundary conditions on the wall for $H_2SO_4$ loss.

To explain an unexpectedly high power dependency of the $H_2SO_4$ diffusion coefficient on temperature observed in our system we accounted in our calculations for involvement of base impurities: dimethyl- (DMA) and trimethyl-amine (TMA). The semi-empirical Fuller formula (Fuller et al., 1966) was used to calculate the diffusion coefficients at dry conditions for solely $H_2SO_4$, and $H_2SO_4$ neutralized with amine bases, namely dimethyamine- and trimethylamine-sulfate. Further, a molecular cluster kinetics model (Olenius et al. 2014) with quantum chemical input data was used to simulate acid–base cluster formation that may lead to the observed behaviour. With the simulations we obtained an effective diffusion coefficient determined in the same way as in the experiments.

The experimental $H_2SO_4$ diffusion coefficients were found to be independent of different initial $[H_2SO_4]$ and a wide range of total flow rates. The values of the diffusion coefficient were found to decrease with increasing relative humidity owing to stronger hydration of $H_2SO_4$ molecules. The observed power dependency of the experimental diffusion coefficients as a function of temperature was found to be of the order of 5.4 when the whole temperature range is accounted for which is in a clear disagreement with predictions from the Fuller method (Fuller et al., 1966)



having a power dependency of 1.75. Since the experimental diffusion coefficients deviate more
from the theory towards the lower temperatures of 278 and 288 K, we suggest that a plausible
explanation for this discrepancy is involvement of impurities such as amines, capable of binding
to acid molecules with the binding strength increasing with decreasing temperature. This
hypothesis is qualitatively supported by clustering kinetics simulations performed using quantum
chemical input data for $H_2SO_4$–dimethylamine and $H_2SO_4$–trimethylamine clusters. Our results
indicate that the effective diffusion coefficient of $H_2SO_4$ in air exhibits a stronger temperature
dependency than predicted from a theory that does not consider cluster formation, and neglecting
this dependency might result in incorrect determination of residual $H_2SO_4$ concentration in
laboratory experiments. More measurements are therefore needed to gain a better understanding
of the temperature dependency of the $H_2SO_4$ diffusion coefficient and the formation of larger
$H_2SO_4$ clusters.

**Acknowledgement**
Authors would like to acknowledge KONE foundation, project CSF No. P209/11/1342, ERC
project 257360-MOCAPAF, and the Academy of Finland Centre of Excellence (project number:
272041) for their financial support.






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



**Table 1.** Summary of simulated and experimental averages (unweighted over RH) of the
effective $H_2SO_4$ diffusion coefficients $D$ ($cm^2$ $s^{-1}$) with standard deviations in parenthesis for
three temperatures 278, 288 and 298 K. The initial base concentration $[base]_{init}$ is set to be either
RH-dependent according to Eq. (3) or RH-independent, and the simulations consider clusters
containing up to one acid and one base molecule ("1×1") or two acid and two base molecules
("2×2") as well as hydrates of the clusters. Power dependencies with respect to the temperature,
obtained as linear fits to the data, are also listed.

| Impurities | | | $D$ (T=278K) | $D$ (T=288K) | $D$ (T=298K) | Power dependency* |
|---|---|---|---|---|---|---|
| Base | $[Base]_{init}$ | Simulated clusters | | | | |
| DMA | (0.1xRH+4) ppt | 1x1 | 0.064 (7%) | 0.069 (7%) | 0.077 (7%) | 2.20/3.18/2.68 |
| | | 2x2 | 0.067 (6%) | 0.072 (6%) | 0.079 (6%) | 2.06/2.92/2.48 |
| | Constant 5 ppt | 1x1 | 0.067 (4%) | 0.072 (4%) | 0.080 (4%) | 2.16/3.01/2.58 |
| | | 2x2 | 0.069 (3%) | 0.074 (4%) | 0.081 (4%) | 2.02/2.77/2.39 |
| TMA | (0.1xRH+2) ppt | 1x1 | 0.065 (7%) | 0.071 (6%) | 0.080 (4%) | 2.40/3.53/2.95 |
| | | 2x2 | 0.068 (6%) | 0.073 (5%) | 0.081 (4%) | 2.16/3.04/2.59 |
| | Constant 2 ppt | 1x1 | 0.065 (2%) | 0.071 (2%) | 0.080 (1%) | 2.41/3.52/2.95 |
| | | 2x2 | 0.067 (2%) | 0.073 (2%) | 0.081 (1%) | 2.15/3.04/2.58 |
| Only SA hydrates | | | 0.079 (4%) | 0.084 (4%) | 0.089 (4%) | 1.67/1.67/1.67 |
| Experiment, this work | | | 0.051 (11%) | 0.055 (11%) | 0.074 (7%) | 2.18/8.70/5.35 |

*power dependency given separately for the temperature ranges 278-288 K / 288-298 K / the
whole dataset temperature range (278-298 K), the same RH range is used for both simulations
and experiment.






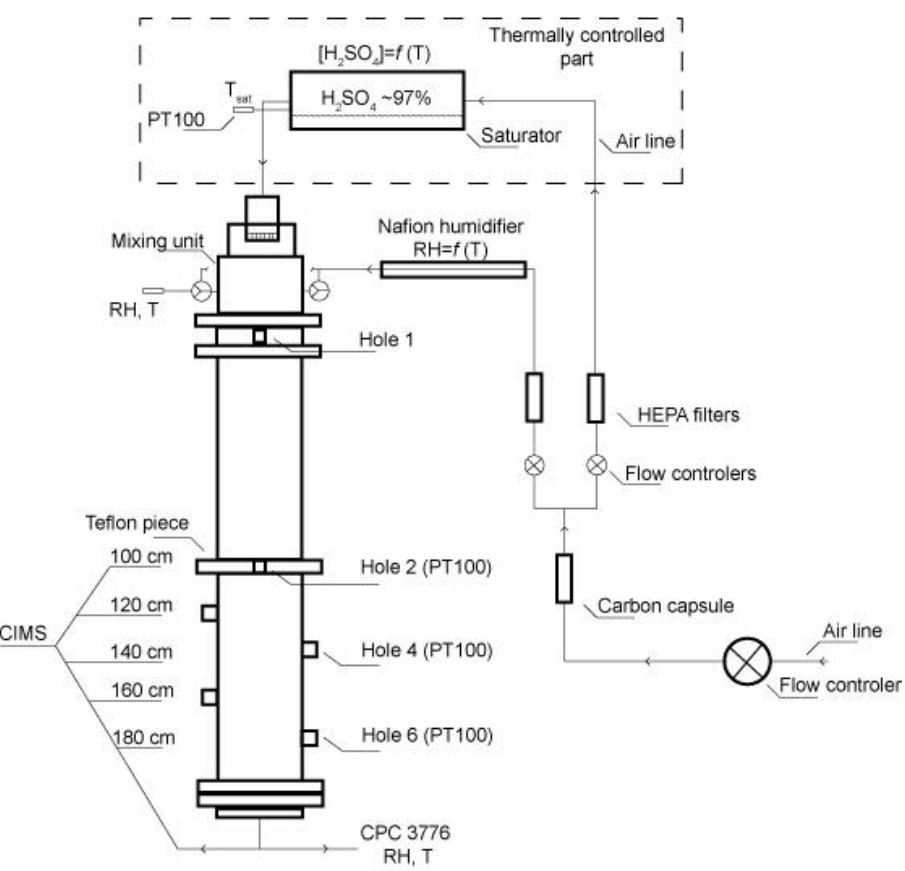



Figure 1. The schematic figure of the FMI flow tube.










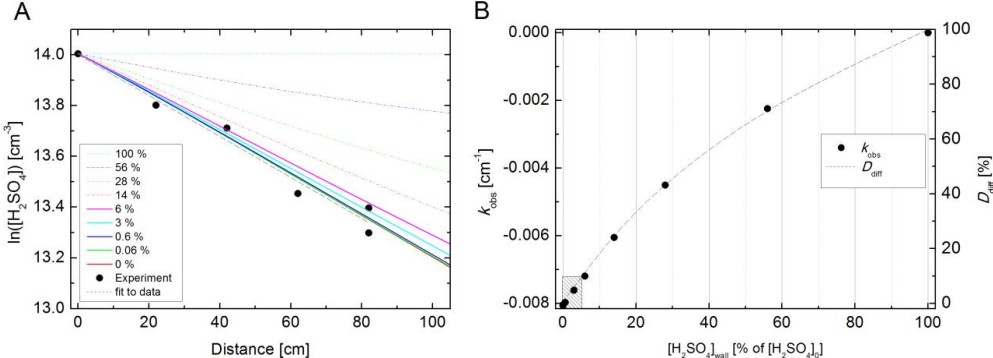


Figure 2. CFD-FLUENT simulations of influence of increased $H_2SO_4$ concentration on the flow
tube wall. A) ln[$H_2SO_4$] as function of distance in the flow tube. B) $k_{obs}$ and diffusion coefficient
difference from the infinite sink boundary condition ($D_{diff}$) as a function of the $H_2SO_4$ wall
concentration expressed as % of initial $H_2SO_4$ concentration, [$H_2SO_4$]$_0$. The simulations
conditions were RH=5 %, $T$=298 K and $Q_{tot}$ =7.6 lpm. When the $H_2SO_4$ concentration on the
wall is ≤ 6 % of [$H_2SO_4$]$_0$, the resulting difference in the obtained diffusion coefficient is within
10 % when compared to diffusion coefficient obtained with infinite sink boundary condition on
the wall, as indicated by the shaded box in bottom left corner in Fig 2B.



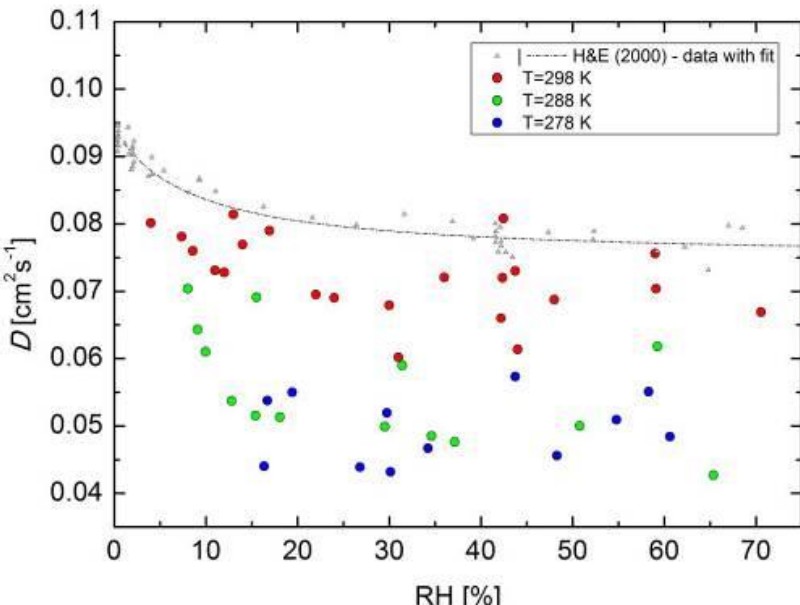


Figure 3. Experimental diffusion coefficient of $H_2SO_4$ in air as a function of relative humidity at
different temperatures compared with fit to the $H_2SO_4$ diffusion in $N_2$ data of Hanson and Eisele

587  (2000).





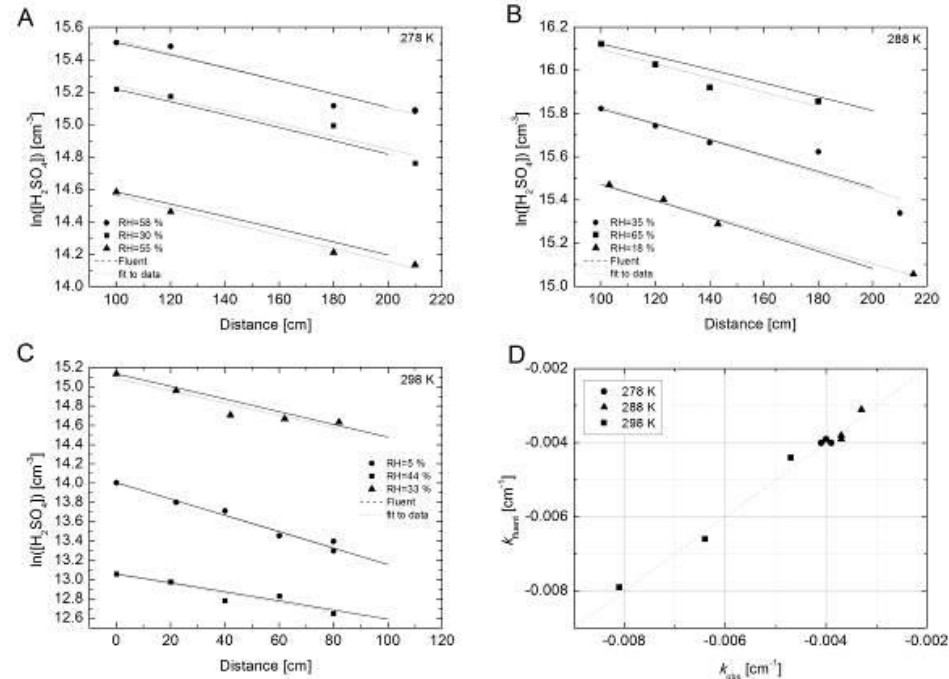


Figure 4. The sulfuric acid losses simulated with CFD-FLUENT model when the experimentally
obtained diffusion coefficients are used as an input at A) $T$=278 K B) $T$=288 K and C) $T$= 298 K.
D) simulated losses rate coefficients compared with experimental values of $k_{obs}$ (cm$^{-1}$) at $T$=278,
288 and 298 K.





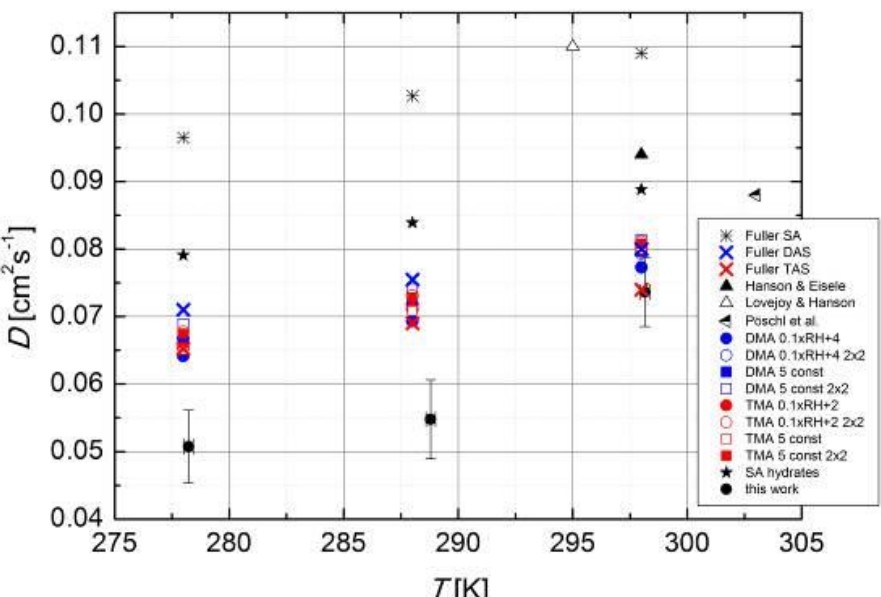


Figure 5. The temperature dependency of the effective $H_2SO_4$ diffusion coefficient, calculated using the Fuller method for dry $H_2SO_4$ (SA), dimethylamine- (DAS) and trimethylamine-sulfate (TAS), both in dry air, data from literature, several assemblies of cluster population simulations (see text for details) and data measured experimentally in this work. The temperature dependency of the experimental diffusion coefficients was found to be a power of 6.

599




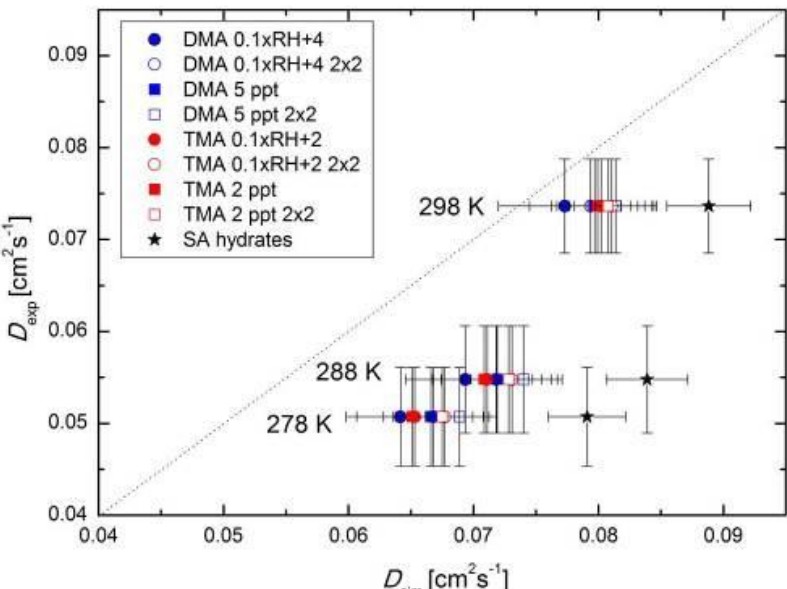

600

Figure 6. Comparison of the experiment and the cluster population simulations at different
temperatures, considered are also different levels and sources of impurities in the system. The
formation of clusters containing up to two $H_2SO_4$ and two base molecules is denoted as "2×2" in
the legend.

605