# Peer review of "Temperature-dependent diffusion coefficient of H2SO4 in air: 2 laboratory measurements using laminar flow technique"

_Atmospheric Chemistry and Physics, 2016_

## Referee Comment (RC1) · Anonymous Referee #1 · 8 Jun 2016

Brus et al. measured the RH and temperature dependence of the diffusion coefficients of gaseous $H_2SO_4$, using a flow tube, and the performance of the experimental set-up is supported by CFD simulations. A much stronger effect of temperature, compared to theories, has been reported, and cluster kinetics modeling has been used to explain and interpret their experimental data. Diffusion coefficients of $H_2SO_4$ are of great importance in atmospheric chemistry, and the results reported by this work are interesting. Nevertheless, I do have a few major concerns which the authors should address before I can recommend this manuscript for final publication:

1) Line 384-386: I do not believe that the CFD simulation can confirm the assumption that the wall of the flow tube is an infinite sink. Actually this is a critical assumption already made in the model (line 166). The modeling result that $[H_2SO_4]$ on the flow tube wall is less than 6% of $[H_2SO_4]_0$ is determined by the spatial resolution used in the model; if a higher resolution is used, $[H_2SO_4]$ on the flow tube wall will be closer to 0. A proper way to confirm this assumption is to measure the diffusion coefficients as a function of pressure in the flow tube, as described in previous work (Fickert et al., 1999; Liu et al., 2009). Though I believe that wall used by Brus et al. should be an infinite sink for $H_2SO_4$, these incorrect statements need to be changed and the proper way to confirm this assumption should be mentioned.

2) I disagree with their proposed temperature dependence of $H_2SO_4$ diffusion coefficients (*D*). Examination of the experimental *D* at different *T* and RH shown in Figure 3 reveals that for the same RH, within the experimental uncertainties there is no significant difference between *D* measured at 278 K and at these measured at 288 K. The very strong dependence of *D* on *T* suggested by the authors are based on three data points at i) 298 K and 4% RH, ii) 288 K and 8% RH, and iii) 278 K and 26% RH (line 290-296). From i) to iii), *T* decreases and RH increases, both very likely leading to the decrease in *D*; in addition, Figure 3 shows that for RH below 15%, RH has a strong effect. One may conclude that instead of temperature, the change of RH can play the major role; therefore, the strong temperature dependence suggested by Brus et al. is not convincing if not wrong.

3) Even if their proposed temperature dependence is correct and can be justified, I feel this study is not complete or does not provide much insight with broad implications. A strong *T* dependence was found and can be explained by clustering with amines. However, the possible presence of amines is an experimental artifact. This manuscript has not yet answered the key question, i.e. the true dependence of $D(H_2SO_4)$ on T, and thus currently may not be suitable for publication by ACP which requires studies with general implications for atmospheric science.

**Reference**

Fickert, S., Adams, J.W., and Crowley, J. N.: Activation of Br2 and BrCl via uptake of HOBr onto aqueous salt solutions, J. Geophys. Res.-Atmos., 104, 23719–23727, 1999.

Liu, Y., Ivanov, A. V., and Molina, M. J.: Temperature dependence of OH diffusion in air and He, Geophys. Res. Lett., 36, L03816, 10.1029/2008gl036170, 2009.

---

## Referee Comment (RC2) · Anonymous Referee #2 · 7 Jul 2016

This paper presents measurements of the wall loss of H2SO4 and its clusters with H2O and perhaps DMA or TMA. There are problems with the interpretation of the data and perhaps in the experimental method as well.

Issues:

1) - Figures are not good enough. Ascertaining data quality is difficult.

- The abstract lists items that are not fully addressed in the paper, such as independence of results upon flow rate (not clearly shown) and H2SO4 level (there seems to be a dependence in Fig. 4 C), or are not factual, such as the claim that clustering with amines explains the temperature dependence (cluster model gives 3 vs. experimental

of 5.4.)

Major issues:

2) How the CIMS is connected to the flow reactor needs to be fully explained. Is the CIMS raised and lowered as it is connected to the different ports? Or are there two elbows in the connecting tube and the CIMS is moved horizontally? Also, and forgive my inattentiveness, is the reported concentration the measured value or that in the flow reactor? Also, it was stated that switching the two flow reactors gave data that validated the technique. Please show this data, and if the temperature data is to remain, at all temperatures. The1 detection of significant concentrations of H2SO4 dimers is important and suggests that there is enough H2SO4 contained in clusters to affect the measurements if these clusters can evaporate. This needs to be more fully explored, i.e., perhaps the general trend that diffusion seems to be 'slow' in this experiment is because dimers or other clusters diffuse more slowly than H2SO4 but then partly evaporate as they travel down the flow reactor. This is possible if indeed there are some amines present that are also lost to the wall, which would tend to lead to decreasing stability of clusters with axial distance.

3) (a) It is not worth much scientifically to use the parabolic velocity - laminar flow equation with the wall loss rate coefficient to get a diffusion coefficient and compare this to a CFD study that assumes a parabolic velocity profile with that diffusion coefficient: it is not too strong a statement to say that NOTHING is validated about the experimental method through that comparison. (b) More importantly, the CFD studies are probably not adequate to the needs of the present study. (i) The gas exiting the mixing region is very likely far from fully developed laminar flow, and (ii) when the flow reactor walls are at temperatures of 10 or 20 C lower than the mixing region walls, there will be buoyancy driven flows, and (iii) the drawing of 7 L/min flow through a small tube is likely to affect the flow patterns and it needs to be shown that this effect on measured H2SO4 does not depend upon which port is being used. It is likely that the details of the mixing region needs to be simulated with a 3D model which would also be required for the

sampling ports.

4) The authors have a lot of work to do to place this data in context, both to motivate readers and to advance the science and their technique. The scatter in the data is large compared to previous measurements, suggesting that the present data can not improve upon previous measurements of the diffusion coefficient. Furthermore, they suggest that amines had influenced their measurements, but this was speculation and really should be somehow verified experimentally (the 'large' dimer concentrations is supporting evidence). The temperature dependency would be something new, but since there are issues with the 298 K data quality and analysis, what can be said about the 288 and 278 K data? Going back to the 298 K data: the relative change with humidity dependence seems to be about the same as that displayed in the earlier data and their should be a curve plotted with their 298 K data using (the previously determined?) equilibrium constants etc. Yet complications linger with uncertainties in whether amines or dimers are significantly affecting the measurements. Are these diffusion measurements of hydrated H2SO4, aminated H2SO4, or a mixture?

---

## Author Comment (AC1) · 14 Oct 2016

We would like to thank both reviewers for their constructive comments on our manuscript, and appreciate pointing out parts needing improvement. Below are our point–by-point answers to the comments.

Anonymous Referee #1

Brus et al. measured the RH and temperature dependence of the diffusion coefficients of gaseous H2SO4, using a flow tube, and the performance of the experimental set-up is supported by CFD simulations. A much stronger effect of temperature, compared to theories, has been reported, and cluster kinetics modeling has been used to explain and interpret their experimental data. Diffusion coefficients of H2SO4 are of great importance in atmospheric chemistry, and the results reported by this work are interesting. Nevertheless, I do have a few major concerns which the authors should address before I can recommend this manuscript for final publication:

1) Line 384-386: I do not believe that the CFD simulation can confirm the assumption that the wall of the flow tube is an infinite sink. Actually this is a critical assumption already made in the model (line 166). The modeling result that [H2SO4] on the flow tube wall is less than 6% of [H2SO4]0 is determined by the spatial resolution used in the model; if a higher resolution is used, [H2SO4] on the flow tube wall will be closer to 0. A proper way to confirm this assumption is to measure the diffusion coefficients as a function of pressure in the flow tube, as described in previous work (Fickert et al., 1999; Liu et al., 2009). Though I believe that wall used by Brus et al. should be an infinite sink for H2SO4, these incorrect statements need to be changed and the proper way to confirm this assumption should be mentioned.

Ad 1) This is probably a misunderstanding and we will reformulate the text to clarify the issue. The mentioned 6 % of initial $H_2SO_4$ concentration on the wall is not a result of the simulations, but a boundary condition for the simulations in which we aimed to assess "how big mistake we could make in determining the diffusion coefficient if the wall is actually *not* acting as an infinite sink". Lines 193-201: we set as a boundary condition for the CFD model the $H_2SO_4$ concentration on the wall to be 0-100% of the initial $H_2SO_4$ concentration. We then investigated the change in the slope (ln([$H_2SO_4$]) vs. distance), and determined the change in the diffusion coefficient accordingly. We found that if [$H_2SO_4$] on the wall is up to 6 % of [$H_2SO_4$]$_0$ we are able to see only 10% change in the determined diffusion coefficient, which is also our experimental uncertainty. In other words, when the wall is emitting up to 6% of [$H_2SO_4$]$_0$, we are not able to recognize it in our experiment.

The method of diffusion coefficient measurements as a function of pressure is not suitable for our study. Generally, in systems including easily nucleating substances like $H_2SO_4$, a small change in the pressure can initiate strong new particle formation, i.e. secondary losses in the system. However, we will also mention other methods for testing the assumption of an infinite wall loss sink in flow tube experiments, as suggested by the reviewer.

2) I disagree with their proposed temperature dependence of H2SO4 diffusion coefficients (D). Examination of the experimental D at different T and RH shown in Figure 3 reveals that for the same RH, within the experimental uncertainties there is no significant difference between D measured at 278 K and at these measured at 288 K. The very strong dependence of D on T suggested by the authors are based on three data points at i) 298 K and 4% RH, ii) 288 K and 8% RH, and iii) 278 K and 26% RH (line 290-296). From i) to iii), T decreases and RH increases, both very likely leading to the decrease in D; in addition, Figure 3 shows that for RH below 15%, RH has a strong effect. One may conclude that instead of temperature, the change of RH can play the major role; therefore, the strong temperature dependence suggested by Brus et al. is not convincing if not wrong.

Ad 2) The suggested temperature dependency is not based on the three data points mentioned by the reviewer, but as stated e.g. in Table 1, as "unweighted averages over RH", which is clarified also on lines 292-294: "The temperature dependency of the experimental diffusion coefficients was found to be a power of 5.4 for the whole dataset and temperature range. "

However, since one might find it incorrect or strange to include averages of the whole dataset, which indeed covers a different RH range for each temperature, we provide new values for the power dependency only for the range of RH that is covered at all three temperatures (15-70%) (please see Table I, last column and last row). The values in Table I in the manuscript will change as follows: the power dependency for the temperature range 278-288 K will change from 2.18 to 1.9, the value for 288-298 K will change from 8.7 to 9.4, and for the whole dataset and temperature range (278-298 K), the change is from 5.35 to 5.56. The manuscript text will change accordingly.

3) Even if their proposed temperature dependence is correct and can be justified, I feel this study is not complete or does not provide much insight with broad implications. A strong T dependence was found and can be explained by clustering with amines. However, the possible presence of amines is an experimental artifact. This manuscript has not yet answered the key question, i.e. the true dependence of D(H2SO4) on T, and thus currently may not be suitable for publication by ACP which requires studies with general implications for atmospheric science.

Ad 3) This manuscript had no ambition to answer the question of the "true dependence of $D(H_2SO_4)$ on T", i.e. the T dependence for the single $H_2SO_4$ molecule not bound to any other molecules, mentioned by the reviewer. We understand that the title might be misleading, and will thus change it to be as follows: Temperature-dependent diffusion of $H_2SO_4$ in air at atmospherically relevant conditions: laboratory measurements using laminar flow technique

First, it is very hard to find an environment in the atmosphere where pure $H_2SO_4$ would exist; even at high altitudes it is likely to be hydrated. As we state on page 3, lines 85-88 : "Such base impurities are unavoidably present also in our experiment, and most probably they originate from the

humidification of the carrier gas (e.g. Benson et al., 2011; Kirkby et al., 2011; Neitola et al. 2015 already cited in the manuscript)."

Within the boundary layer, impurities like ammonia or amines are always present (e.g. Ge et al. 2011). The ranges of $H_2SO_4$ concentrations, RH, and temperature used in this study, as well as the concentration of impurities, represent typical ambient values and are atmospherically relevant. Thus we consider our study to be suitable for general implications in atmospheric science. We will also add the reference to Ge et al. (2011), and will add some further discussion on the evidence that in components that bound to sulfuric acid are likely present in any natural environments (e.g. Petäjä et al. (2011) already cited in the manuscript).

References

Fickert, S., Adams, J.W., and Crowley, J. N.: Activation of Br2 and BrCl via uptake of HOBr onto aqueous salt solutions, J. Geophys. Res.-Atmos., 104, 23719–23727, 1999.

Liu, Y., Ivanov, A. V., and Molina, M. J.: Temperature dependence of OH diffusion in air and He, Geophys. Res. Lett., 36, L03816, 10.1029/2008gl036170, 2009.

Ge, X., A. S. Wexler, S. L. Clegg (2011), Atmospheric amines – Part I. A review, *Atmos. Env.*, 45, 524-546, doi:dx.doi.org/10.1016/j.atmosenv.2010.10.012

---

## Author Comment (AC2) · 14 Oct 2016

We would like to thank both reviewers for their constructive comments on our manuscript, and appreciate pointing out parts needing improvement. Below are our point–by-point answers to the comments.

Anonymous Referee #2

This paper presents measurements of the wall loss of H2SO4 and its clusters with H2O and perhaps DMA or TMA. There are problems with the interpretation of the data and perhaps in the experimental method as well.

Issues: 1) - Figures are not good enough. Ascertaining data quality is difficult. - The abstract lists items that are not fully addressed in the paper, such as independence of results upon flow rate (not clearly shown) and H2SO4 level (there seems to be a dependence in Fig. 4 C), or are not factual, such as the claim that clustering with amines explains the temperature dependence (cluster model gives 3 vs. experimental 5.4.)

Ad 1) The quality of figures suffered due to conversion to pdf format, and we apologize for that. Our goal was to keep the manuscript compact and not to present every test of proper operation of the flow tube. Since the reviewer pointed out that this may seem inadequate, we will create Supplementary Materials for the current manuscript and address all performed tests there. The independence of the wall loss rate coefficient on flow rate is presented in Fig S1. The dependence or independence on $H_2SO_4$ concentration cannot be judged from Fig 4C, since Fig 4C represents measurements at different $H_2SO_4$ concentrations and also at different RHs; the experiments performed at a lower RH have a steeper slope. The independence on $H_2SO_4$ level, as well as data obtained when we switched the flow tube parts are now together covered in Fig S2.

The authors do *not* claim that clustering with amines fully explains the temperature dependence, nor was it argued anywhere within the manuscript. Instead, we *suggest* that the strong temperature dependence of the observed diffusion coefficient might be explained by increased diffusion volume of $H_2SO_4$ molecules due to stronger clustering with base impurities like amines at lower temperatures. The clustering kinetics simulations using quantum chemical data available for $H_2SO_4$-$H_2O$, $H_2SO_4$-DMA-$H_2O$, and $H_2SO_4$-TMA-$H_2O$ systems were performed in order to test this hypothesis. Naturally also the simulation approach involves uncertainties in the quantitative rate constants. Nevertheless, the simulation results for the measurable diffusion coefficient qualitatively agree with the experimental data, also in terms of the relative differences in the diffusion coefficient for the three temperature points, thus supporting the hypothesis on the effect of amines. But the important message that can be drawn from our results is: Even at conditions with sulfuric acid concentrations similar to the real atmosphere and with concentrations of impurities likely lower than in any natural environments, the impurities still affect the effective sulfuric acid diffusion coefficient and its temperature dependence. We will discuss this issue further in the revised manuscript.

[Figure]

Fig S1. Measured wall loss rate coefficient as a function of total flow in the flow tube, at three RH 5, 14 and 40 %. Horizontal lines (separately for each RH) represent the mean, and the vertical error bars the standard deviation.

Major issues: 2) How the CIMS is connected to the flow reactor needs to be fully explained. Is the CIMS raised and lowered as it is connected to the different ports? Or are there two elbows in the connecting tube and the CIMS is moved horizontally? Also, and forgive my inattentiveness, is the reported concentration the measured value or that in the flow reactor? Also, it was stated that switching the two flow reactors gave data that validated the technique. Please show this data, and if the temperature data is to remain, at all temperatures. The detection of significant concentrations of H2SO4 dimers is important and suggests that there is enough H2SO4 contained in clusters to affect the measurements if these clusters can evaporate. This needs to be more fully explored, i.e., perhaps the general trend that diffusion seems to be 'slow' in this experiment is because dimers or other clusters diffuse more slowly than H2SO4 but then partly evaporate as they travel down the flow reactor. This is possible if indeed there are some amines present that are also lost to the wall, which would tend to lead to decreasing stability of clusters with axial distance.

Ad 2) The inlet of the CIMS is about the mid-height of the flow tube (110 cm from ground), the sampling pipe (100 + 22 cm elbow) thus covers whole necessary length for sampling from top to bottom of the flow tube. The CIMS' rack is equipped with wheels allowing movement, CIMS was moved in horizontal direction of maximum distance of about 1 meter from the flow tube when the sampling pipe is connected to mid-height port of the flow tube.

All mentioned concentrations of $H_2SO_4$ through the manuscript are measured values, which will be clarified in the revised version of the manuscript, and as mentioned on lines 134-136: "In this study the actual $H_2SO_4$ concentrations are not of particular interest, we focus here only on the relative loss of $H_2SO_4$ along the flow tube". However, the concentration in the flow tube can be obtained by multiplying by a wall loss factor (WLF) of approximately 4, determined by Brus et al. (2011).

Experimental data obtained when we switched the two parts of the flow reactor are now provided in Fig. S2. Data are available only for T=298 K, and the obtained wall loss rate coefficients $k_w$ ($s^{-1}$) at each RH are within the experimental uncertainty of 10%, even though different total flows and different $H_2SO_4$ concentrations were used. For lower temperatures (288 and 278 K) we used only a setup where the flow tube part with sampling ports was at lower position (100 - 200 cm). This will be clarified in the revised version of the manuscript.

In our experiment we observed a very slight increase in the [dimer]/[monomer] ratio as a function of residence time (distance), and a slight decrease or no correlations with residence time for solely dimers at all three temperatures; please see panels A and B in Fig. S3. This implies that the loss rate of monomers is higher than the loss rate of dimers. Dimers are lost to walls or they evaporate. Since amines are known to have a stabilizing effect on $H_2SO_4$ clusters, we assess that the wall loss is the dominating loss process. However, one has to be very cautious to draw any solid conclusions out of Fig. S3. The provided figure should be taken only as qualitative. The assessment concerning dimers is also qualitatively supported by the cluster kinetics simulations: including the formation of dimers somewhat increases the apparent diffusion coefficient instead of decreasing it (Fig. 6), as the aminated dimers are bound tightly enough to not decompose in significant amounts before detection.

[Figure]

| RH[%] | $k_{obs}$[cm⁻¹] | $k_w$[s⁻¹] | $Q_{tot}$[lmin⁻¹] |
|---|---|---|---|
| 5 | -0.0069 | 0.036 | 8.9 |
|  | -0.0063 | 0.033 | 8.8 |
| 14 | -0.0075 | 0.033 | 7.5 |
|  | -0.0099 | 0.035 | 6 |
| 40 | -0.0064 | 0.030 | 8 |
|  | -0.0081 | 0.029 | 6 |

Fig S2. Measured $H_2SO_4$ concentration as a function of distance in the flow tube at 298 K, at three RHs 5, 14 and 40 %, different levels of $H_2SO_4$ concentrations, and when the flow tube part with sampling ports is in the first half (0-100 cm, denoted as Part_1) or in the second half (100-200 cm, denoted as Part_2). The resulting wall loss rate coefficients $k_w$ from the fits are presented in the table together with total flows $Q_{tot}$ in the flow tube.

[Figure]

Fig S3. A) Measured dimer to monomer ratio and B) dimer concentration, both panels at several different levels of RH and at three temperatures 278, 288 and 298 K, as a function of residence time spent in the flow tube.

3) (a) It is not worth much scientifically to use the parabolic velocity - laminar flow equation with the wall loss rate coefficient to get a diffusion coefficient and compare this to a CFD study that assumes a

parabolic velocity profile with that diffusion coefficient: it is not too strong a statement to say that NOTHING is validated about the experimental method through that comparison.

(b) More importantly, the CFD studies are probably not adequate to the needs of the present study. (i) The gas exiting the mixing region is very likely far from fully developed laminar flow, and (ii) when the flow reactor walls are at temperatures of 10 or 20 C lower than the mixing region walls, there will be buoyancy driven flows, and (iii) the drawing of 7 L/min flow through a small tube is likely to affect the flow patterns and it needs to be shown that this effect on measured H2SO4 does not depend upon which port is being used. It is likely that the details of the mixing region needs to be simulated with a 3D model which would also be required for the sampling ports.

Ad 3a) The CFD modelling is performed only to verify the proper operation of the flow tube, and to validate the assumptions made in the analysis. We think it is worth to show that the experimentally obtained diffusion coefficient is producing the same wall loss rate coefficient when inserted to the CFD model under the same assumptions, i.e. a laminar flow. Assuming any other type of flow profile inside the flow tube than a laminar one does not seem justifiable when the Reynold's number (Re) is about 160 within our setup.

To demonstrate the change in the losses inside the flow tube compared to our approach, we made the following set of CFD simulations for all three temperatures and a constant mid-range RH≈30%: a) we used the diffusion coefficient obtained from the experiment, but with a constant (flat-plug type) flow profile as initial condition, b) we used diffusion coefficients obtained from eq. (3) in Hanson and Eisele (2000) with the temperature dependency of $\sim T^{1.75}$ obtained from literature, and an initial parabolic flow profile. The simulations are summarized in Fig S4. Clearly, none of the two alternative approaches is able to reproduce the measured loss profile.

Ad 3b) The mixing is taking place in the mixer which is -18 cm from the beginning of the flow tube. This may not be very clear from Fig. 1, and we will update the manuscript accordingly. The distance ($X_L$) needed for the flow to become fully developed can be approximated by $X_L = 0.03 \times Re \times d$, where the Reynolds number Re is about 160 in our experiments, and the diameter d is 6 cm. The fully developed laminar profile could be found approx. 12 cm from the beginning of the flow tube. Generally, we try to minimize the temperature gradient in a way that the saturator temperature and the flow tube temperature are the same (isothermal conditions, see Fig. 1), and the amount of sulfuric acid in the system is then governed by the amount of flow through the saturator and the subsequent mixing flow. The only part that is not temperature-controlled is the mixer, but it is insulated. We want the mixer to be slightly warmer to avoid any condensation and particle formation in it. However, with the experiments taking relatively long (hours, one profile) the mixer will cool down a bit (at 278 and 288 K), and so the temperature gradient is not prominent. In experiments conducted at 298 K we switched the two parts of the flow tube and obtained satisfactorily reproducible data; please see Fig S2. To ensure no disturbance in the flow pattern due to slight

temperature gradients in experiments conducted at lower temperatures (278 and 298 K), we used only the setup where the sampling ports were at the lower part of the flow tube (100-200 cm); please see Fig. 4 A and B in the manuscript. This will be clarified in a revised version of the manuscript to avoid any confusion. The sampling ports have a zigzag (left-right) configuration. If there was any problem with flow pattern inside the tube, it would be visible in the experimental data: the obtained points on one or the other side of the flow tube would be systematically lower or higher, and it would be clearly visible if the flow tube was badly assembled or leaking.

[Figure]

Fig S4. Simulated loss rate coefficients $k$_fluent (cm$^{-1}$) compared with experimental values $k$_obs (cm$^{-1}$) with three different CFD setups at $T$=278, 288 and 298 K. The dotted 1:1 line denotes the perfect agreement. Black squares: The measured sulfuric acid losses. Blue solid lines: Losses simulated with CFD-FLUENT model when a constant (flat-plug type) flow profile and the experimentally obtained diffusion coefficients are used as an initial condition.   Red solid lines: Losses simulated with diffusion coefficients obtained from eq. (3) in Hanson and Eisele (2000) and a parabolic flow profile. Black solid lines: Losses simulated with a parabolic flow profile and the experimentally obtained diffusion coefficients.  Black dashed lines: A fit to the experimental data. .

4) The authors have a lot of work to do to place this data in context, both to motivate readers and to advance the science and their technique. The scatter in the data is large compared to previous measurements, suggesting that the present data cannot improve upon previous measurements of the diffusion coefficient. Furthermore, they suggest that amines had influenced their measurements, but this was speculation and really should be somehow verified experimentally (the 'large' dimer concentrations is supporting evidence). The temperature dependency would be something new, but since there are issues with the 298 K data quality and analysis, what can be said about the 288 and 278 K data? Going back to the 298 K data: the relative change with humidity dependence seems to be about the same as that displayed in the earlier data and their should be a curve plotted with their 298 K data using (the previously determined?) equilibrium constants etc. Yet complications linger with uncertainties in whether amines or dimers are significantly affecting the measurements. Are these diffusion measurements of hydrated $H_2SO_4$, aminated $H_2SO_4$, or a mixture?

Ad 4) The presence of amines and their concentration was repeatedly measured in our system as stated in the manuscript on page 8, lines 236-238. We are going to provide the measured values within the Supplementary materials as follows:

Concentrations of impurities in our system were measured from the source –the saturator- with MARGA (Monitor for Aerosols and Gas in Ambient Air system; *Makkonen et al.*, 2012) with improved detection of amines by a quadrupole mass spectrometer (MS, Shimadzu LCMS-2020). MS was operated with electrospray ionization with positive mode. For quantitative analysis deuterated diethyl-d10-amine (Sigma-Aldrich: Isotec™; Sigma-Aldrich, St. Louis, MO, USA) was used as an internal standard together with 3-point external calibration for all amines. Amines were detected as their $M^{+1}$ ions and therefore impact of impurities with the same retention times but different molecular masses was removed. The limits of detection (LODs) were calculated as three times the standard deviation of the blank levels for di-methyl-amine (DMA) and tri-methyl-amine (TMA) and from signal-to-noise ratio (3:1) for mono-methyl-amine (MMA). LODs were 7 ppt for MMA, 1.7 ppt for DMA and 0.3 ppt for TMA.

Table ST1.The concentration of amines in *ppt* measured at three temperatures (278, 288 and 298 K) in the saturator when purified air is used as carrier gas, all at dry conditions. MMA concentrations are provided even though they are below LOD, i.e. in practice zero.

|  | MMA | std(MMA) | DMA | std(DMA) | TMA | std(TMA) |
|---|---|---|---|---|---|---|
| 278 K | 1.92 | 0.69 | 3.14 | 1.16 | 1.17 | 0.06 |
| 288 K | - | - | 2.95 | 1.20 | 1.38 | 0.07 |
| 298K | 2.12 | 0.51 | 2.73 | 0.72 | 1.31 | 0.07 |

The elevated dimer-to-monomer ratios is another indirect evidence for the presence of stabilizing species. There is no doubt that presence of amines will influence the processes in the flow tube via strong acid-base reactions.

To plot a curve similar to the previous study by Hanson and Eisele (2000) would be a good exercise for a different molecular system, but for the present study not very helpful. The parameters obtained from the fit to our experimental values would in practice have no physical meaning. Hanson and Eisele (2000) used the hydrate theory to obtain equilibrium constants for two $H_2SO_4$ hydrates from the fit, however assuming only sulfuric acid and water interactions. This is not relevant to our case, as we know that both water and amines are present. For this complex system of clustering species, the number of free parameters determining the cluster distribution is much larger, and obtaining a set of equilibrium constants is not possible by simple fitting. This is why we have performed clustering kinetics simulations using available quantum chemical data. This is currently the most reliable approach to assess the effects of hydration and acid-base clustering on the measurement results.

While the complexity of the system as well as impossibility to fully determine the nature of impurities prevents us providing the equilibrium coefficients for the system, our work still provides an important message for the atmospheric science community: at atmospheric sulfuric acid concentrations with impurity concentrations likely lower than in any natural environments, the clustering with impurities still significantly lowers the effective sulfuric acid diffusion coefficient and changes its expected temperature dependence - because clustering itself also depends on temperature. Due to the importance of our findings for atmospheric modelling we expect these issues will be more closely looked in year to come.

References

Brus, D., Neitola, K., Hyvärinen, A.-P., Petäjä, T., Vanhanen, J., Sipilä, M., Paasonen, P., Kulmala, M., and Lihavainen, H.: Homogenous nucleation of sulfuric acid and water at close to atmospherically relevant conditions, Atmos. Chem. Phys., 11, 5277– 5287, doi:10.5194/acp-11-442 5277-2011, 2011.

Hanson, D. R. and Eisele, F. L.: Diffusion of H2SO4 in humidified nitrogen: Hydrated H2SO4, J. Phys. Chem. A, 104, 1715–1719, 2000.

Makkonen, U., Virkkula, A., Mäntykenttä, J., Hakola, H., Keronen, P., Vakkari, V., and Aalto, P. P.: Semi-continuous gas and inorganic aerosol measurements at a Finnish urban site: comparisons with filters, nitrogen in aerosol and gas phases, and aerosol acidity, Atmos. Chem. Phys., 12, 5617–5631, doi:10.5194/acp-12- 5617-2012, 2012